# Neurobiological Theories of Addiction: A Comprehensive Review

Carmen Ferrer-Pérez [1], Sandra Montagud-Romero [2] and María Carmen Blanco-Gandía [3,*]

1 Department of Developmental Psychology, University of Valencia, Av. Blasco Ibáñez 21, 46010 Valencia, Spain; carmen.ferrer-perez@uv.es

2 Department of Psychobiology, University of Valencia, Av. Blasco Ibáñez 21, 46010 Valencia, Spain; sandra.montagud@uv.es

3 Department of Psychology and Sociology, University of Zaragoza, C/Ciudad Escolar s/n, 44003 Teruel, Spain

* Correspondence: mcblancogandia@unizar.es

**Abstract:** It is essential to develop theories and models that enable us to understand addiction's genesis and maintenance, providing a theoretical and empirical framework for designing more effective interventions. Numerous clinical and preclinical research studies have investigated the various brain and physiological mechanisms involved in addictive behavior. Some researchers have gone a step further, developing what we may refer to as "neurobiological theories of addiction", which are scientific models that can explain and predict different addiction phenomena. Many of these neurobiological theories are not mutually exclusive but rather extensions and refinements of earlier theories. They all share a similar definition of addiction as a chronic disease characterized by a loss of control over substance consumption, with the brain being identified as the principal organ involved. Most propose a multifactorial causation in which both biological and environmental factors interact, accentuating or causing neurobiological dysfunction in structures and brain circuits involved in behavior and motivation. This review delves into primary neurobiological theories of addiction, commencing with the opponent-process theory—one of the earliest comprehensive explanations of the addictive process. Subsequently, we explore more contemporary formulations connecting behavioral alterations in the addictive process to changes and disruptions in various brain systems.

**Keywords:** addiction; neurobiology; opponent process; dopamine; incentive sensitization; allostasis; compulsion; neuroinflammation





## 1. Introduction

Drug addiction is characterized by a loss of control over drug-seeking and -consumption, despite the profound negative consequences this has on the individual's life [1]. While the acute effects of a substance depend on its psychoactive properties, the progression of addiction converges into a series of problems that are common and severely impact all spheres of the individual's life, compromising interpersonal, economic, and health status [2]. Thus, in chronic drug users it is common to present several physical problems including brain damage and atrophy, circulatory system issues, premature aging, among others. From a socio-economic perspective, common problems include homelessness, criminal behavior, unemployment, social isolation, and dependence [3].

From all of these devastating consequences arises the necessity to develop preventive strategies against the initiation of drug consumption and to develop effective therapeutic interventions for patients suffering from addiction. For this purpose, the first necessary step involves the development of theoretical models that provide an explanation of the genesis of and maintenance of the addictive process, allowing the formulation of a theoretical framework upon which to base therapeutic approaches.

The classical and lay perception of addiction considers it as a problem of lack of willpower, where individuals with an addictive disorder simply would not "try hard enough" to cease substance consumption. Fortunately, preclinical and clinical scientific

research has revealed that, although the manifestation of addiction and its consequences are unique to each individual, many of the behavioral effects of drugs are due to alterations in the brain that are similar across individuals [4].

Different researchers have focused on unraveling which brain structures and mechanisms are involved in addiction, e.g., [5–11]. Some researchers have gone a step further, developing what we may refer to as "neurobiological theories of addiction", which are scientific models that can explain and predict different addiction phenomena.

Many of these neurobiological theories are not mutually exclusive but rather extensions and refinements of earlier theories. They all share a similar definition of addiction as a chronic disease characterized by a loss of control over substance consumption, with the brain being identified as the principal organ involved [1,12,13]. Most propose a multifactorial causation in which both biological and environmental factors interact, accentuating or causing neurobiological dysfunction in structures and brain circuits involved in behavior and motivation [1,10–13]. Finally, these theories assert that, after repeated contact with addictive substances, learning and brain changes occur. These changes would affect cognitive, emotional, and behavioral spheres and may persist despite prolonged abstinence, explaining the chronicity of the disorder [12,14].

This review delves into primary neurobiological theories of addiction, commencing with the opponent-process theory [15], one of the earliest comprehensive explanations of the addictive process. Subsequently, we explore more contemporary formulations connecting behavioral alterations in the addictive process to changes and disruptions in various brain systems. While this review emphasizes addiction as a brain disease, it is crucial to consider that addressing addiction requires a holistic approach. This involves considering not only neurobiological variables but also other determinants, including socioeconomic and psychological factors. However, we believe that the current review allows for an understanding of the evolution of the concept of addiction as a disease with an organic substrate and how new treatments for this disease have been developed.

## 2. Opponent-Process Theory—Solomon and Corbit (1974)

The opponent-process theory was developed by Solomon and Corbit [15] as a novel explanatory paradigm for human affective and motivational processes. According to the authors, this theory also allows for an understanding of phenomena related to addiction and aversion.

The theory posits that when a hedonically positive affective response (primary process) is activated in the brain, a series of mechanisms simultaneously initiate a hedonically opposite response (opponent process). The purpose of this opponent response is to counteract the activation produced by the primary response and restore the initial state of homeostasis. Solomon and Corbit [15] further assert that repeated activation of a primary process reinforces the duration and intensity of its opponent process. Finally, they conclude their theory by stating that both the primary and opponent processes are susceptible to Pavlovian conditioning.

According to these authors, drug consumption acutely induces a primary process associated with gratification and pleasure, leading to a heightened sense of wellbeing. Simultaneously, it triggers an opponent process aimed at restoring physiological and brain functions to their original state. This opponent process is characterized by states that contrast with those acutely induced by the drug, resulting in a response marked by unpleasant sensations, both psychological (e.g., irritability or depression) and physical (sweating, thirst, among others). Furthermore, both the primary and the opponent processes become conditioned to various cues linked to drug consumption, such as paraphernalia or contextual cues. Exposure to these conditioned stimuli will evoke both the drug-associated primary process and its opponent process.

Continuing with Solomon and Corbit's explanation [15], sustained drug consumption promotes an increase in the intensity and duration of the opponent process. This phenomenon has several consequences for an individual's response to a drug.

Firstly, there is the development of tolerance or habituation to the rewarding and pleasurable effects of the drug. This occurs because the pleasurable primary process activated after substance consumption is rapidly counteracted by an increasingly intense opponent process.

Secondly, since the primary process does not extend its duration but the opponent process does, a state of prolonged discomfort associated with the opponent response emerges after drug consumption. This state, appearing more rapidly with repeated drug use, is directly identified by the authors as withdrawal syndrome.

Thirdly, environmental and contextual stimuli conditioned to the drug elicit both processes, but as the opponent process is strengthened relative to the primary one, the individual's experience becomes hedonically negative. Therefore, contextual cues would elicit a response that is similar to withdrawal symptoms. Consequently, contextual cues evoke a response akin to withdrawal symptoms, which may promote the need to seek and consume the drug (craving) to alleviate the discomfort caused by this activation of the opponent process.

Therefore, after repeated drug consumption, the primary process is counteracted by an increasingly intense and enduring opponent process, leading to the perpetuation of drug consumption to counteract its negative effects. This theory provides a general explanation for the development of tolerance, the appearance of withdrawal symptoms, craving, and the establishment of a consumption spiral aimed at alleviating these effects.

This theory, which is on the verge of celebrating its 50th anniversary, is considered a milestone in the study of human affective processes. It has also served as a foundation for the development of more modern addiction theories, such as Koob and LeMoal's "Allostasis" theory [16]. However, the opponent-process theory falls short in explaining individual variability in addiction vulnerability, that is, why not all individuals with repeated exposure to a drug develop addiction. Moreover, it does not make any reference to the involved brain mechanisms.

### 3. Dopaminergic Hypothesis of Addiction—Wise (1980)

A significant portion of both animal and human behavior is governed by mechanisms of positive reinforcement. The initial steps to identify the underlying brain circuits of reward were taken by Olds and Milner [17]. These researchers found that rats acquired the behavior of pressing a lever only when this action resulted in intracranial stimulation of certain brain regions, such as the medial forebrain bundle and the mesolimbic cortical dopaminergic pathway, referred to as "pleasure centers" by the researchers. They asserted that these "pleasure" regions would be the same as those activated by the satisfaction of basic needs such as food, thirst, and sex, explaining the reinforcement–maintenance of these behaviors. In the subsequent years, numerous intracranial electrical stimulation studies were conducted to "map" the different pleasure centers in the brain [18].

Concurrently, pharmacological studies were conducted to determine which neurotransmitter was mediating reward signaling in these circuits. Initially, it was believed that noradrenaline pathways controlled reward, as the administration of noradrenergic antagonists decreased the acquisition of intracranial self-stimulation behavior [19]. However, it was later established that this effect was due to the drowsiness and impairment caused by the blockade of this neurotransmitter rather than an effect on the reward response [20]. As soon as the first selective dopamine antagonists were developed, it was confirmed that their administration had a specific effect on reward. Wise and collaborators studied the pharmacodynamics of drugs such as cocaine and amphetamine, discovering that both increased dopamine in the mesolimbic dopaminergic system, while dopamine antagonists were capable of blocking the rewarding effects of these psychoactive drugs [21,22]. These authors also confirmed that other drugs like alcohol, benzodiazepines, and barbiturates produced their reinforcement through a similar mechanism, disinhibiting dopamine neurons rather than directly exciting them [20]. Finally, they demonstrated that dopaminergic antagonists blocked natural reward associated with food [23], reinforcing the theory

that natural rewards, intracranial self-stimulation reward, and drugs act on a common dopamine-dependent substrate.

This substrate corresponds to the brain's reward system, which aligns with the mesolimbic cortical dopaminergic pathway. This pathway includes projections originating from the ventral tegmental area and terminating in the nucleus accumbens, with additional connections to other structures such as the hippocampus, prefrontal cortex, amygdala, olfactory tubercle, and lateral septal nucleus (Figure 1). Subsequent studies confirmed that the reward level induced by a drug is directly related to the phasic increase in dopamine levels in the nucleus accumbens [24].

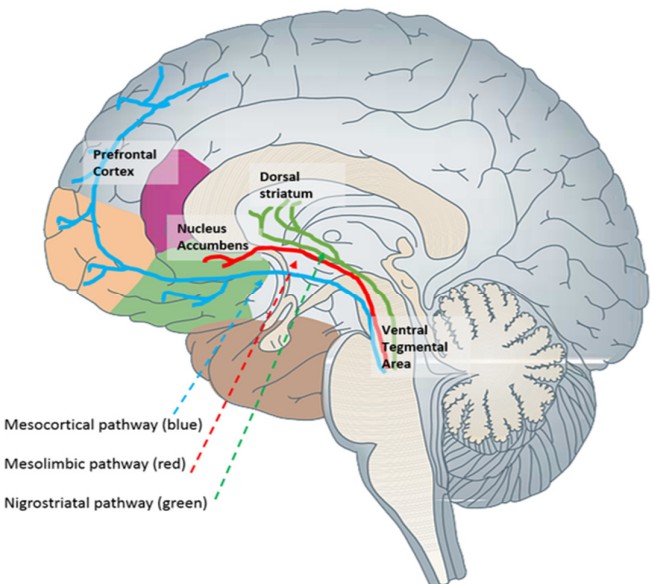

**Figure 1.** Main dopaminergic pathways. The brain reward system is primarily associated with the mesolimbic (red) and mesocortical (blue) pathways.

According to the "Dopaminergic Hypothesis of Addiction" drugs act through a common mechanism of increasing dopamine in the brain's reward system, promoting positive reinforcement, and motivating drug consumption and addiction [25]. In addition to an explanatory model of the mechanism of action of all drugs, this theory provides a preliminary neurophysiological understanding of the phenomenon of vulnerability to addiction.

Numerous preclinical and clinical studies supported that low dopamine release was a predictor of vulnerability to addiction, while an adequate dopaminergic tone played a role in the resilience of individuals who would not escalate in drug consumption [25]. In the same vein, it was also described that the level of expression of D2/D3 dopamine receptors in the striatum modulated the subjective response to different substances [26] Individuals with fewer dopamine receptors, with no prior drug use, reported greater pleasure or "liking" after the consumption of psychostimulants [26,27]. Animal studies showed that these individuals engaged in more intense drug self-administration behaviors [28]. Therefore, dopaminergic hypofunction in the mesolimbic system would, according to this theory, be a risk factor by enhancing the euphoric response and reward obtained from drugs.

However, this theory, by exclusively considering drug-induced positive reinforcement in its explanation, lacks explanatory value for various phenomena in the addictive process, such as withdrawal syndrome or craving, which would be better elucidated using a negative-reinforcement mechanism. Subsequently, based on this theory, others emerged asserting that the role of dopamine was not solely to mediate the experience of euphoria and reward but also to promote the formation of motivational salience, the establishment of habits, and reward expectations associated with cues [29]. These new insights into the functions of dopamine led to the development of new neurobiological theories of addiction, such as Robinson and Berridge's "Incentive Sensitization Theory" [29].

#### 4. Incentive Sensitization Theory—Robinson and Berridge (1993)

Drugs of abuse induce a highly intense and prolonged activation of the reward system, with dopamine increases between 3 and 10 times greater than those observed with natural rewards [30]. After repeated consumption, elevations beyond "natural levels" will lead to alterations in the functioning of the reward system.

As mentioned earlier, dopamine's role in this system extends beyond encoding reward; it also participates in the memory consolidation, habit formation, and motivational salience associated with the drug. Therefore, neuroadaptations in this system will compromise all these processes. In this regard, Robinson and Berridge [29] suggested that neuroadaptive changes following repeated consumption make individuals hypersensitive to environmental cues associated with drug reward. These authors state that the reward system becomes sensitized; therefore, in the presence of environmental cues associated with the drug, individuals are involuntarily motivated to invest resources and energy in approaching the substance, which is perceived as relevant or necessary. This incentive salience could be understood as an attentional bias toward stimuli associated with drug consumption, highly charged with the power to motivate consummatory behavior. Therefore, these authors conclude that after repeated use, motivation shifts from an initial state of pleasure-driven consumption (liking) to pathological craving (wanting), reflecting a process of sensitization and conditioning of the reward system.

This theory is very interesting as it enables the explanation of craving, drug seeking, and drug consumption when the addict is exposed to environments and cues associated with drug use. However, it is limited since the authors have not delved into the neuroadaptive processes that would explain this hypersensitivity to cues, nor have they connected these processes with a greater or lesser susceptibility to addiction development.

#### 5. Habit and Compulsion Theory—Robbins and Everitt (1999); Everitt and Robbins (2005)

The central idea of the "Habit and Compulsion" theory is that, in the addictive process, the initial substance consumption occurs voluntarily, driven by its recreational effects (positive reinforcement). However, with repeated consumption, the individual progressively loses control over the consumption behavior, which turns into a compulsive behavior (stimulus response) that is difficult to extinguish.

Building on the incentive sensitization theory [29], Robbins and Everitt [31] acknowledge that repeated overactivation of the dopaminergic system would lead to alterations that would cause drugs to acquire a high-incentive motivational value, that stimulates drug craving in the presence of associated cues [29]. However, they assert that this theory would not explain why addicts find it impossible to control consumption behavior and why they persist despite the severe consequences of their addiction. Everitt and Robbins address this limitation by stating that neuroplastic processes also impact dopaminergic circuits controlling goal-directed behaviors and habit formation.

Thus, in the transition from voluntary and occasional consumption to compulsive consumption, a progressive shift in the locus of control of drug-associated behaviors would be observed, moving from top-down control to a regulation of the behavior controlled by the basal ganglia. Behavior would cease to be controlled by ventral striatal regions, extensively connected with the prefrontal cortex and amygdala, to be controlled by dorsal striatal regions, specialized in the maintenance of motor sequences. Therefore, the prefrontal cortex would increasingly have less inhibitory control over drug-associated motor behaviors, which would become compulsive and disinhibited.

The activation threshold necessary to initiate these motor habits would be progressively reduced, making the exposure to environmental cues enough to trigger drug-seeking and -consumption behaviors. Robbins and Everitt [31] assert that this loss of control would not occur in all individuals who repeatedly consume a drug; only a percentage of vulnerable individuals would progress to compulsive consumption. Through studies with animal models, they found that rodents exhibiting more impulsive behavior robustly acquired drug self-administration behavior. In fact, in these preclinic studies researchers found that

impulsive animals continued performing drug-seeking behaviors even when contextual cues indicated that the drug was not present or that an electric shock (punishment) would be administered if they pressed the lever [32,33]. For these authors, impulsivity is the underlying factor of the susceptibility to escalate drug consumption, relapse, and ultimately lose control over substance use. Other researchers have conducted clinical studies confirming that the activation threshold required to initiate these motor habits would gradually decrease, making exposure to environmental cues sufficient enough to trigger drug-seeking and -consumption behaviors. According to Robbins and Everitt [31], this loss of control would not occur in all individuals who repeatedly consume a drug; only a subset of vulnerable individuals would progress to compulsive consumption. Their research using animal models revealed that rodents displaying more impulsive behavior robustly acquired drug self-administration behavior. Notably, in these preclinical studies, impulsive animals persisted in drug-seeking behaviors even when contextual cues indicated the absence of the drug or the administration of an electric shock (punishment) upon lever pressing [32]. For these authors, impulsivity is the underlying factor of the susceptibility to escalating drug consumption, relapse, and eventual loss of control over substance use. Additional clinical studies conducted by other researchers have supported that the construct of impulsivity is a predisposing factor associated with vulnerability to substance-use disorders, and also it is a consequence of chronic consumption [34].

### 6. Allostasis Theory of Addiction—Koob and Le Moal (1997)

The "Allostasis" theory, developed by Koob and Le Moal [16] shares with previous theories the idea that in the initial phases of the addictive process, consumption is motivated by the expectation of positive reinforcement. However, after intense and chronic consumption, behavior maintenance should be attributed to a process of negative reinforcement, as only the drug can alleviate the activation and discomfort that occurs during periods of abstinence. To explain this transition, Koob and Le Moal [16,35] formulated a theory that integrates the foundations of the opponent-process theory [15] and the concept of homeostasis.

Living organisms seek to actively regulate and maintain a stable internal environment, allowing them to adapt to changes in the external environment (homeostasis). As previously discussed, the consumption of drugs and their abuse leads to an activation of the reward system above its "natural" levels, making this aberrant activation a threat to the homeostasis of this system. This triggers the activation of two corrective mechanisms that aim to counteract the drug's effects: on one hand, there is a loss of function in the reward system, resulting in an increase in its activation thresholds; on the other hand, there is hyperactivation of stress or anti-reward systems.

Koob and Le Moal [16] assert that chronic activation of these corrective mechanisms can strengthen them to the point of counteracting and masking the rewarding effects of drugs. In fact, these corrective mechanisms would strengthen to such an extent that they would produce an overcompensation beyond the initial homeostatic level. Therefore, a new set point or "allostasis" would be established, as an attempt of anticipatory compensation for future drug consumption.

While the establishment of this new set point is an adaptive response aimed at anticipating and favoring the stability of the reward system, it has a negative impact on individual affect and motivation. This is because the person undergoes a chronic state of reward-system hypofunction and a state of stress-system hyperactivity. In other words, a constant state of stress and anti-reward dominates the subject's motivational balance. In this state, basal dopaminergic tone and even natural rewards (such as food or sex) prove insufficient to return the system to its natural level. Only drugs abused will be capable of offsetting the negative consequences of this allostatic state, leading to their consumption not for their reinforcing effects but for their ability to alleviate this dysphoric state during abstinence (see Figure 2).

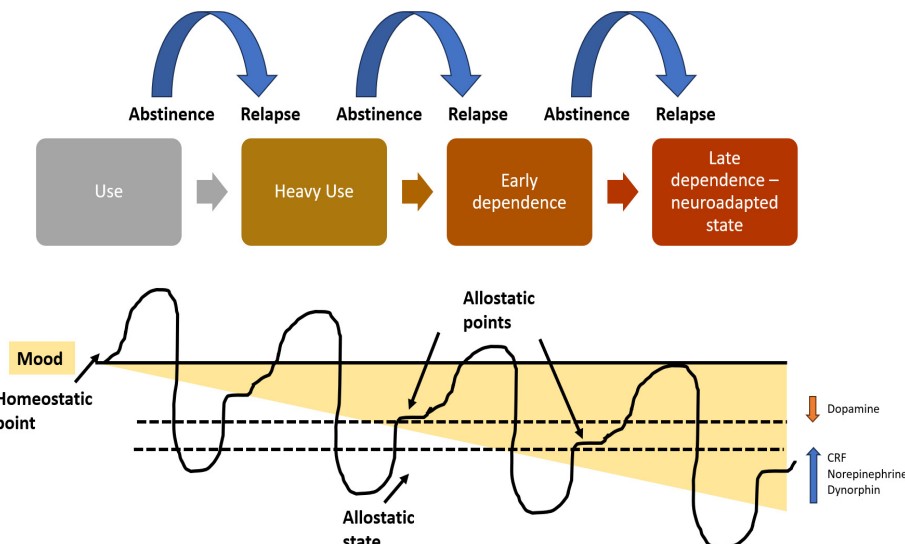

**Figure 2.** Diagram of the progression of drug-consumption motivation from a state of positive reinforcement to the maintenance of consumption for relief of discomfort (negative reinforcement). Adapted from Koob and Schulkin [35].

This theory is the first to include structures outside the reward system to explain dysphoria and craving during abstinence. According to the authors, neuroadaptations within the reward system would underlie the diminished pleasurable effects of the drug. Additionally, they propose that the hyperactivity of the brain stress system, along with the subsequent increase in its neuroendocrine products (CRF and noradrenaline), accounts for anxiety and discomfort states during abstinence. This theory is the first to incorporate structures outside the reward system to elucidate dysphoria and craving during abstinence. According to the authors, neuroadaptations within the reward system underlie the diminished pleasurable effects of the drug.

Ultimately, the Allostasis Theory of addiction would help in understanding why stress acts as a risk factor in all stages of the addictive process, being a key factor in relapse. As previously mentioned, the neuroendocrine stress response actively participates in modulating the reward system, generating an anti-reward response that would amplify the rewarding value associated with the drug. In this sense, individuals that are more susceptible to stress, and those presenting a more pronounced and lasting endocrine response, would be particularly vulnerable to these negative effects.

## 7. New Approaches to the Study of the Addictive Process: Neuroinflammation

In recent decades, there has been a remarkable increase in our understanding of the neurobiological mechanisms involved in addiction. Currently, it is acknowledged that, beyond dopamine and stress hormones, other neurotransmission systems, such as the endocannabinoid and oxytocin systems, play a crucial role in understanding vulnerability and the progression to addiction in certain individuals. In addition, the role of the glutamatergic system must not be overlooked, as it enhances neuronal excitability and modulates neuroplasticity [5]. Many of the alterations observed in addiction-related behaviors are commonly attributed to changes in excitatory signaling and the maintenance of glutamate homeostasis [36–38]. In this context, scientists have focused on studying specific mechanisms and therapeutic targets in detail, rather than generating new comprehensive neurobiological theories addressing addiction.

One of the systems recently linked to the genesis and progression of addiction is the immune system. Growing evidence suggests the influence of the immune system in the onset of various mental illnesses, including mood disorders, anxiety, autism, schizophrenia, and addiction, e.g., [39–43]. The central nervous system can receive and process signals from the immune system. An illustrative example of this intercommunication is "sickness

behavior", characterized by social avoidance, apathy, and anhedonia in response to an infectious process. In reality, these symptoms constitute components of an adaptive response aimed at conserving energy to enhance survival, given that the defensive actions of the immune system, such as fever or antibody production, are resource-intensive for our body [44].

The signals mediating the appearance of this sickness behavior are proinflammatory cytokines, protein substances released by immune-system cells in response to a pathogen. These substances serve to coordinate the defensive response against the pathogen, promoting the arrival of other immune cells to the infection site and favoring inflammation. At some point, these cytokines can cross the blood–brain barrier, thus entering the central nervous system. There, microglia, resident immune cells of the brain, upon detecting these cytokines, activate and initiate an inflammatory response by releasing cytokines themselves [45]. The presence of cytokines in the brain (neuroinflammation) is the central signal that coordinates the sickness behavior response and initiates the behavioral alterations observed in individuals experiencing illness.

In this sense, it has been described that a chronic inflammatory state, which would induce a state of chronic sickness behavior, are a risk factor for the development of mood disorders. For example, patients with inflammatory diseases such as psoriasis or rheumatoid arthritis have a higher risk of developing depression throughout their lives [46].

Many drugs abused have a high inflammatory potential and the ability to alter the functioning of the immune system [47–50]. The inflammatory potential of alcohol at the hepatic level is well known. In the brain, alcohol is a xenobiotic, a substance foreign to our body, so the microglia is activated in its presence [51]. In response to the presence of ethanol, brain immune cells initiate an inflammatory cascade. If alcohol consumption is persistent, a state of constant neuroinflammation in the brain will occur, causing atrophy, neurodegeneration, and neuronal death. Furthermore, neuronal damage and death will further activate brain immune cells, which will attempt to purge dead tissues, further exacerbating inflammation and its negative consequences. This neuroinflammatory potential would not be exclusive to ethanol as it is shared by most drugs abused.

Neuroinflammatory processes allow us to explain the emotional distress experienced by many addicts after chronic consumption as a consequence of a drug-induced state of brain inflammation. Moreover, these inflammatory processes caused by chronic consumption lead to atrophy and neuronal death, explaining part of the cognitive impairment that occurs in long-term drug consumers.

This theory would also explain how stress increases vulnerability to drug use. Stress experiences would also have the potential to induce an inflammatory response both in the periphery and within the brain. This stress-induced inflammatory response can increase the response to drug reward through its action on the hormonal axis that is activated during stress response [52]. Therefore, cytokines can enhance the activity of the stress axis by facilitating the release of CRF, subsequently directly amplifying dopaminergic function in the reward circuit (see Figure 3).

In situations of chronic inflammation, either due to chronic stress or repeated drug use, the induction of stress hormones will produce neuroadaptations that increase the hedonic response to different drugs of abuse [54–56]. Finally, this theory would help to explain part of the individual vulnerability to addiction. There are individuals whose immune systems are much more sensitive and reactive to threats, whether from pathogens or other types of triggers. These individuals would react by deploying an exaggerated inflammatory response after drug consumption or exposure to stress. This heightened response would promote, as mentioned before, greater release of stress hormones with their consequent action on the reward system, as well as greater neuronal damage. Therefore, it would be expected that these individuals whom are more susceptible to inflammation would also be more susceptible to sensitize their reward system and experience greater cognitive impairment after drug consumption. Table 1 provides an overview of the primary

contributions and limitations of the Neuroinflammation Theory regarding addiction, along with the other neurobiological theories discussed in this review.

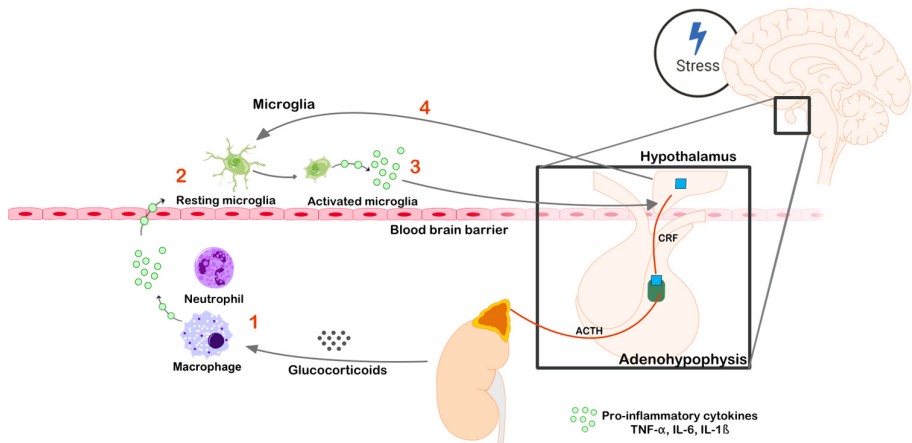

**Figure 3.** Relationship between the stress hormone axis and the immune system. (**1**) The repetitive release of glucocorticoids following chronic stress induces a primed state in peripheral immune cells, enhancing their response to subsequent immune challenges; (**2**) Activation of stress-primed immune cells can lead to an exaggerated pro-inflammatory response, eventually reaching the brain and promoting the activation of resident immune cells (neuroinflammation); (**3**) Activated microglia release pro-inflammatory cytokines that impact the stress axis, thereby promoting the release of corticotropin-releasing factor (CRF); (**4**) Norepinephrine signaling induced by stress can directly activate microglia, inducing a pro-inflammatory state in these immune cells. Adapted from Ferrer-Pérez et al. [53].

**Table 1.** Main contributions and limitations of the classical and new neurobiological theories of addiction.

| Theory | Contributions | Limitations |
|---|---|---|
| **Opponent-Process Theory (Solomon and Corbit, 1974)** | The theory provides a comprehensive explanation addressing both the initial pleasure phase and the subsequent aversive phase in the addictive process. It offers an understanding of the persistence of emotional effects over time, explaining the challenge of overcoming addiction. Recognizes the importance of time and repetition in the development and maintenance of addiction, highlighting the dynamic nature of the process. It proposes concepts such as tolerance and habituation. It can be applied to various substances and addictive behaviors, providing a solid theoretical foundation for understanding different types of addictions. | The theory might oversimplify the complexity of psychological and neurobiological processes involved in addiction. The theory focuses more on emotional response processes than on the underlying causes of addiction. The theory may be perceived as a unidirectional approach by primarily emphasizing the oppositional process after the initial pleasure, without considering other intervening factors. |
| **Dopaminergic Hypothesis of Addiction (Wise, 1980)** | The hypothesis highlights the central role of dopamine in brain circuits associated with reward, providing a foundation for understanding the hedonic component of addiction. It provides a clear and specific explanation of how dopamine influences reward and reinforcement processes. It can be applied to a wide variety of addictive substances, offering a useful theoretical framework for understanding addiction to different drugs. The hypothesis has received support through studies demonstrating changes in dopamine levels in relation to the administration of addictive substances and reward-seeking behaviors. | By primarily focusing on dopamine, the hypothesis may oversimplify the complexity of neurochemical and behavioral factors involved in addiction. It does not adequately address psychosocial and contextual factors that also play crucial roles in the development and maintenance of addiction. Dopaminergic response can vary significantly among individuals, suggesting that factors beyond dopamine may be equally important. Although dopamine is implicated in reward, the hypothesis cannot always accurately predict the development and course of addiction in all cases. |

**Table 1.** *Cont.*

| Theory | Contributions | Limitations |
|---|---|---|
| **Incentive Sensitization Theory (Robinson and Berridge, 1993)** | The theory emphasizes the significance of sensitization to stimuli associated with drugs, providing insight into how motivation for seeking addictive substances develops. It offers an explanation of how sensitization can persist over time, contributing to an understanding of the chronic nature of addiction. The theory incorporates the motivational dimension, highlighting the transition from initial pleasure-seeking to motivated and persistent drug seeking. Applicable to various addictive substances, providing a broad theoretical framework that is applicable to different types of addictions. | The theory centers more on motivation for seeking addictive substances than addressing the underlying causes of addiction, potentially limiting overall understanding. The theory may not adequately address relapse processes and factors contributing to relapse in addiction. |
| **Habit and Compulsion Theory (Robbins and Everitt, 1999; Everitt and Robbins, 2005)** | The theory focuses on brain circuits and neuronal structures associated with habits and compulsions, providing a solid neurobiological foundation. Adaptability to different types of addictions: applicable to a variety of substances and addictive behaviors, providing a broad and applicable theoretical framework. The theory considers the influence of the environment and learning in the formation and persistence of habits and compulsions, enriching the perspective. | By focusing on habits and compulsions, the theory might oversimplify the diversity of factors involved in different types of addictions. This theory would not explain why addicts find it impossible to control consumption behavior and why they persist despite the severe consequences of their addiction. It may not comprehensively address the initial motivations leading to addiction, focusing more on the later phases of the addictive cycle. While considering the environment, the theory might lack specificity in precisely how environmental factors influence the formation of habits and compulsions. |
| **Allostasis Theory of Addiction (Koob and Le Moal, 1997)** | The theory centers on allostatic processes, providing a perspective that highlights the continuous adaptation of the neurobiological system in response to drug-related demands. Offers an explanation of long-term neurobiological changes associated with addiction, addressing the need to comprehend dynamics over time. Considers stress and stress factors as significant elements in addiction, expanding the understanding beyond substances themselves. The theory addresses the importance of allostatic processes in the relapse cycle, providing a comprehensive view of addictive processes. | Although it considers stress, it might not comprehensively address psychosocial and environmental factors that are also crucial in addiction. By strongly emphasizing neurobiological aspects, the theory might overlook some important psychological and social aspects of addiction. The response to the neuroadaptive mechanisms can vary significantly among individuals. |
| **New Approaches to the Study of the Addictive Process: Neuroinflammation** | Research on neuroinflammation represents an emerging approach in understanding addiction, providing new insights into underlying biological processes. Addresses the interaction between the immune system and the central nervous system, potentially enriching the understanding of addiction beyond traditional aspects. Research on neuroinflammation could lead to the identification of new therapeutic targets for addiction treatment, expanding available options. Understanding neuroinflammation could have significant implications for addressing comorbidities associated with addiction and neuropsychiatric disorders. | The interaction between neuroinflammation and addiction is complex, making it challenging to identify clear causal relationships and specific mechanisms. While ongoing research exists, solid empirical evidence regarding the precise contribution of neuroinflammation to addiction may still be limited. Some neuroinflammation approaches may focus on later phases of the addictive process, leaving gaps in understanding initial events and predisposition. The clinical application of neuroinflammation research may face challenges, from identifying effective interventions to practical implementation in clinical settings. |

## 8. Conclusions

The exploration of diverse neurobiological theories has significantly enhanced our understanding of the intricate phenomena underlying drug addiction. These theories, ranging from classic paradigms like the opponent-process theory to more contemporary perspectives such as the Allostasis Theory and neuroinflammation, have collectively contributed to unraveling the complexities of addictive processes.

Each theory or hypothesis is focused on one or a few aspects related to the addictive process. The opponent-process theory highlights the emotional duality of pleasure and aversion, providing a framework for understanding the persistence of addiction. However, the Dopaminergic Hypothesis focuses on dopamine as pivotal in reward and reinforcement, emphasizing its central role. The Incentive Sensitization Theory emphasizes sensitization to stimuli and the transition to motivated drug-seeking behavior while the Habit and Compulsion Theory underscores the formation of drug-related habits and compulsions. The Allostasis Theory highlights the continuous adaptation of the neurobiological system, focusing on long-term changes. Additionally, new approaches, with an emphasis on neuroinflammation, represent a cutting-edge understanding of immunological factors in addiction.

It is essential to critically address the limitations of these theories, such as the lack of predictive accuracy and oversimplification of underlying complexity. Many theories tend to oversimplify the complexity of addiction by focusing on a limited set of neurobiological factors. Addiction is a multifaceted phenomenon involving complex interactions among biological, psychological, and social factors. Secondly, while various neurobiological mechanisms associated with addiction have been identified, the lack of specificity in causal processes and the complex interactions between them make it challenging to formulate precise interventions. In addition, responses to drugs and vulnerability to addiction vary significantly among individuals. Neurobiological theories often cannot fully explain this variability and may overlook genetic and epigenetic factors. Furthermore, the clinical application of neurobiological theories can be challenging. Despite advances in scientific understanding, the effective translation of this knowledge into clinical treatments remains an area under development; as well as this, it is a challenge to predict the development and course of addiction in individual cases.

Further research is needed to validate and refine these theories, as well as to explore new avenues that may provide an even more comprehensive understanding of addiction and guide more effective interventions in the future. Future perspectives include integrating these theories for a more holistic approach, considering interactions between different systems; these could provide scientists with the ability to develop personalized therapies based on understanding the specific mechanisms of each individual.

**Author Contributions:** Conceptualization, C.F.-P.; writing—original draft preparation, C.F.-P., S.M.-R. and M.C.B.-G.; writing—review and editing, S.M.-R. and M.C.B.-G.; supervision, C.F.-P. All authors have read and agreed to the published version of the manuscript.

**Funding:** This research received no external funding.

**Conflicts of Interest:** The authors declare no conflicts of interest.

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
