# Peer review of "Neurobiological Theories of Addiction: A Comprehensive Review"

_psychoactives, doi:10.3390/psychoactives3010003_

Round 1
Reviewer 1 Report
Comments and Suggestions for Authors
In this manuscript, the authors reviewed primary neurobiological theories of addiction. Although interesting and of potential interest for the readers, I think that the manuscript requires some revisions to get it acceptable for publication:
- - The abstract and the introduction are exactly the same. I suggest the authors to rewrite these sections. The abstract should summarize the manuscript while the introduction should provide background information and set the context of the review.
- - The conclusion section is confusing and needs to be rewritten. The authors should summarize the main points of the review, but also providing future perspectives and implications/recommendations as well as critical inputs.
- - I should suggest the authors to include a table summarizing the pros and cons of each neurobiological theory.
-
Comments on the Quality of English Language
The English is good. It requires only few corrections.
Author Response
Please review the responses in the attached PDF document. Thank you for your review efforts and suggestions for improvement.

Reviewer 2 Report
Comments and Suggestions for Authors
In the present studies, the authors described different theories for drug addiction based upon their neurobiological changes. They discussed different period of the theories according to the findings in different periods. The manuscript was well written. I have a minor consideration.
Glutamate and its receptors play critical roles in the brain and also contribute to the drug addiction. As we know, glutamate is the King of neurotransmitter in the brain. Therefore, it's role in the drug addiction can not be neglected. The author should mention this important neurotransmitter in the drug addiction.
Author Response
Please review the responses in the attached PDF document. Thank you for your review efforts and suggestions for improvement

Reviewer 3 Report
Comments and Suggestions for Authors
The manuscript submitted for review is an analysis of the neurobiological theories of addiction. In general, the topic is highly actual, the manuscript is of use and should be published after the revision.
Comments:
The small number of citations of works from the last 5 years is rather surprising (out of 40 bibliographic items, only 1 has been published after 2018). I believe that the topic has been developing vigorously over the past 5 years, and focusing on old articles is not the best approach to reviewing it. Therefore, I am convinced that the authors must carry out a detailed literature review and supplement the citations with references to the latest research in this field. Additionally, the novelty of the review in comparison with the previous reports should be clearly shown.
I also encourage authors to describe clearly the methodology of their research.
I recommend publication after major revisions.
Author Response

(The authors gave the same response as above.)

Round 2
Reviewer 1 Report
Comments and Suggestions for Authors
The authors adequately addressed all my concerns. I think that the manuscript is now suitable for publication.
Comments on the Quality of English Language
The authors adequately addressed all my concerns. This version of the manuscript is now suitable for publication.